# Diving into the shallows: a computational perspective on large-scale shallow learning

**Siyuan Ma**      **Mikhail Belkin**
Department of Computer Science and Engineering
The Ohio State University
{masi, mbelkin}@cse.ohio-state.edu

## Abstract

Remarkable recent success of deep neural networks has not been easy to analyze theoretically. It has been particularly hard to disentangle relative significance of architecture and optimization in achieving accurate classification on large datasets. On the flip side, shallow methods (such as kernel methods) have encountered obstacles in scaling to large data, despite excellent performance on smaller datasets, and extensive theoretical analysis. Practical methods, such as variants of gradient descent used so successfully in deep learning, seem to perform below par when applied to kernel methods. This difficulty has sometimes been attributed to the limitations of shallow architecture.

In this paper we identify a basic limitation in gradient descent-based optimization methods when used in conjunctions with smooth kernels. Our analysis demonstrates that only a vanishingly small fraction of the function space is *reachable* after a polynomial number of gradient descent iterations. That drastically limits the approximating power of gradient descent leading to over-regularization. The issue is purely algorithmic, persisting even in the limit of infinite data.

To address this shortcoming in practice, we introduce EigenPro iteration, a simple and direct preconditioning scheme using a small number of approximately computed eigenvectors. It can also be viewed as learning a kernel optimized for gradient descent. Injecting this small, computationally inexpensive and SGD-compatible, amount of approximate second-order information leads to major improvements in convergence. For large data, this leads to a significant performance boost over the state-of-the-art kernel methods. In particular, we are able to match or improve the results reported in the literature at a small fraction of their computational budget. For complete version of this paper see https://arxiv.org/abs/1703.10622.

## 1 Introduction

In recent years we have witnessed remarkable advances in many areas of artificial intelligence. Much of this progress has been due to machine learning methods, notably deep neural networks, applied to very large datasets. These networks are typically trained using variants of stochastic gradient descent (SGD), allowing training on large data with modern GPU hardware. Despite intense recent research and significant progress on SGD and deep architectures, it has not been easy to understand the underlying causes of that success. Broadly speaking, it can be attributed to (a) the structure of the function space represented by the network or (b) the properties of the optimization algorithms used. While these two aspects of learning are intertwined, they are distinct and may be disentangled.

As learning in deep neural networks is still largely resistant to theoretical analysis, progress can be made by exploring the limits of shallow methods on large datasets. Shallow methods, such as kernel methods, are a subject of an extensive and diverse literature, both theoretical and practical. In particular, kernel machines are universal learners, capable of learning nearly arbitrary functions given a sufficient number of examples [STC04, SC08]. Still, while kernel methods are easily implementable and show state-of-the-art performance on smaller datasets (see [CK11, HAS+14,

DXH$^+$14, LML$^+$14, MGL$^+$17] for some comparisons with DNN's) there has been significantly less progress in applying these methods to large modern data. The goal of this work is to make a step toward understanding the subtle interplay between architecture and optimization and to take practical steps to improve performance of kernel methods on large data.

The paper consists of two main parts. First, we identify a basic underlying limitation in using gradient descent-based methods in conjunction with smooth (infinitely differentiable) kernels typically used in machine learning, showing that only very smooth functions can be approximated after polynomially many steps of gradient descent. This phenomenon is a result of fast spectral decay of smooth kernels and can be readily understood in terms of the spectral structure of the gradient descent operator in the least square regression/classification setting, which is the focus of our discussion. Slow convergence leads to severe over-regularization (over-smoothing) and suboptimal approximation for less smooth functions, which are arguably very common in practice, at least in the classification setting, where we expect fast transitions near the class boundaries.

This shortcoming of gradient descent is purely algorithmic and is not related to the sample complexity of the data. It is also not an intrinsic flaw of the kernel architecture, which is capable of approximating arbitrary functions but potentially requiring a very large number of gradient descent steps. The issue is particularly serious for large data, where direct second order methods cannot be used due to the computational constraints. While many approximate second-order methods are available, they rely on low-rank approximations and, as we discuss below, lead to over-regularization (approximation bias).

In the second part of the paper we propose EigenPro iteration (see http://www.github.com/EigenPro for the code), a direct and simple method to alleviate slow convergence resulting from fast eigen-decay for kernel (and covariance) matrices. EigenPro is a preconditioning scheme based on approximately computing a small number of top eigenvectors to modify the spectrum of these matrices. It can also be viewed as constructing a new kernel, specifically optimized for gradient descent. While EigenPro uses approximate second-order information, it is only employed to modify first-order gradient descent, leading to the same mathematical solution as gradient descent (without introducing a bias). EigenPro is also fully compatible with SGD, using a low-rank preconditioner with a low overhead per iteration. We analyze the step size in the SGD setting and provide a range of experimental results for different kernels and parameter settings showing five to 30-fold acceleration over the standard methods, such as Pegasos [SSSSC11]. For large data, when the computational budget is limited, that acceleration translates into significantly improved accuracy. In particular, we are able to improve or match the state-of-the-art results reported for large datasets in the kernel literature with only a small fraction of their computational budget.

## 2 Gradient descent for shallow methods

**Shallow methods.** In the context of this paper, shallow methods denote the family of algorithms consisting of a (linear or non-linear) *feature map* $\phi : \mathbb{R}^{\mathbb{N}} \to \mathcal{H}$ to a (finite or infinite-dimensional) Hilbert space $\mathcal{H}$ followed by a linear regression/classification algorithm. This is a simple yet powerful setting amenable to theoretical analysis. In particular, it includes the class of kernel methods, where $\mathcal{H}$ is a Reproducing Kernel Hilbert Space (RKHS).

**Linear regression.** Consider $n$ labeled data points $\{(\mathbf{x}_1, y_1), ..., (\mathbf{x}_n, y_n) \in \mathcal{H} \times \mathbb{R}\}$. To simplify the notation let us assume that the feature map has already been applied to the data, i.e., $\mathbf{x}_i = \phi(\mathbf{z}_i)$. Least square linear regression aims to recover the parameter vector $\alpha^*$ that minimize the empirical loss such that $\alpha^* = \arg\min_{\alpha \in \mathcal{H}} L(\alpha)$ where $L(\alpha) \stackrel{\text{def}}{=} \frac{1}{n} \sum_{i=1}^n (\langle \alpha, \mathbf{x}_i \rangle_{\mathcal{H}} - y_i)^2$. When $\alpha^*$ is not uniquely defined, we can choose the smallest norm solution.

Minimizing the empirical loss is related to solving a linear system of equations. Define the data matrix $X \stackrel{\text{def}}{=} (\mathbf{x}_1, ..., \mathbf{x}_n)^T$ and the label vector $\mathbf{y} \stackrel{\text{def}}{=} (y_1, ..., y_n)^T$, as well as the (non-centralized) covariance matrix/operator, $H \stackrel{\text{def}}{=} \frac{1}{n} \sum_{i=1}^n \mathbf{x}_i \mathbf{x}_i^T$. Rewrite the loss as $L(\alpha) = \frac{1}{n} \|X\alpha - \mathbf{y}\|_2^2$. Since $\nabla L(\alpha) \mid_{\alpha = \alpha^*} = 0$, minimizing $L(\alpha)$ is equivalent to solving the linear system

$$H\alpha - \mathbf{b} = 0 \tag{1}$$

with $\mathbf{b} = X^T \mathbf{y}$. When $d = \dim(\mathcal{H}) < \infty$, the time complexity of solving the linear system in Eq. 1 directly (using Gaussian elimination or other methods typically employed in practice) is $O(d^3)$. For kernel methods we frequently have $d = \infty$. Instead of solving Eq. 1, one solves the dual $n \times n$ system $K\alpha - \mathbf{y} = 0$ where $K \stackrel{\text{def}}{=} [k(\mathbf{z}_i, \mathbf{z}_j)]_{i,j=1,...,n}$ is the kernel matrix . The solution can be written as $\sum_{i=1}^n k(\mathbf{z}_i, \cdot)\alpha(\mathbf{z}_i)$. A direct solution would require $O(n^3)$ operations.

**Gradient descent (GD).** While linear systems of equations can be solved by direct methods, such as Gaussian elimination, their computational demands make them impractical for large data. Gradient descent-type methods potentially require a small number of $O(n^2)$ matrix-vector multiplications, a much more manageable task. Moreover, these methods can typically be used in a stochastic setting, reducing computational requirements and allowing for efficient GPU implementations. These schemes are adopted in popular kernel methods implementations such as NORMA [KSW04], SDCA [HCL$^+$08], Pegasos [SSSSC11], and DSGD [DXH$^+$14]. For linear systems of equations gradient descent takes a simple form known as the Richardson iteration [Ric11]. It is given by

$$\alpha^{(t+1)} = \alpha^{(t)} - \eta(H\alpha^{(t)} - \mathbf{b}) \tag{2}$$

It is easy to see that for convergence of $\alpha^t$ to $\alpha^*$ as $t \to \infty$ we need to ensure that $\|I - \eta H\| < 1$, and hence $0 < \eta < 2/\lambda_1(H)$. The explicit formula is

$$\alpha^{(t+1)} - \alpha^* = (I - \eta H)^t(\alpha^{(1)} - \alpha^*) \tag{3}$$

We can now describe the *computational reach* of gradient descent $\mathcal{CR}_t$, i.e. the set of vectors which can be $\epsilon$-approximated by gradient descent after $t$ steps, $\mathcal{CR}_t(\epsilon) \stackrel{\text{def}}{=} \{\mathbf{v} \in \mathcal{H}, s.t. \|(I - \eta H)^t \mathbf{v}\| < \epsilon \|\mathbf{v}\|\}$. It is important to note that any $\alpha^* \notin \mathcal{CR}_t(\epsilon)$ cannot be $\epsilon$-approximated by gradient descent in less than $t + 1$ iterations. Note that we typically care about the quality of the solution $\|H\alpha^{(t)} - \mathbf{b}\|$, rather than the error estimating the parameter vector $\|\alpha^{(t)} - \alpha^*\|$ which is reflected in the definition. We will assume that the initialization $\alpha^{(1)} = 0$. Choosing a different starting point does not change the analysis unless second order information is incorporated in the initialization conditions.

To get a better idea of the space $\mathcal{CR}_t(\epsilon)$ consider the eigendecomposition of $H$. Let $\lambda_1 \geq \lambda_2 \geq \ldots$ be its eigenvalues and $\mathbf{e}_1, \mathbf{e}_2, \ldots$ the corresponding eigenvectors/eigenfunctions. We have $H = \sum \lambda_i \mathbf{e}_i \mathbf{e}_i^T$. Writing Eq. 3 in terms of eigendirection yields $\alpha^{(t+1)} - \alpha^* = \sum(1 - \eta\lambda_i)^t\langle\mathbf{e}_i, \alpha^{(1)} - \alpha^*\rangle\mathbf{e}_i$. Hence putting $a_i \stackrel{\text{def}}{=} \langle\mathbf{e}_i, \mathbf{v}\rangle$ gives $\mathcal{CR}_t(\epsilon) = \{\mathbf{v}, s.t. \sum(1 - \eta\lambda_i)^{2t}a_i^2 < \epsilon^2 \|\mathbf{v}\|^2\}$. Recalling that $\eta < 2/\lambda_1$ and using the fact that $(1 - 1/z)^z \approx 1/e$, we see that a necessary condition for $\mathbf{v} \in \mathcal{CR}_t$ is $\frac{1}{3}\sum_{i,s.t.\lambda_i < \frac{\lambda_1}{2t}} a_i^2 < \sum_i(1 - \eta\lambda_i)^{2t}a_i^2 < \epsilon^2 \|\mathbf{v}\|^2$. This is a convenient characterization, we will denote $\mathcal{CR}'_t(\epsilon) \stackrel{\text{def}}{=} \{\mathbf{v}, s.t. \sum_{i,s.t.\lambda_i < \frac{\lambda_1}{2t}} a_i^2 < \epsilon^2 \|\mathbf{v}\|^2\} \supset \mathcal{CR}_t(\epsilon)$. Another convenient but less precise necessary condition for $\mathbf{v} \in \mathcal{CR}_t$ is that $\left|(1 - 2\lambda_i/\lambda_1)^t \langle\mathbf{e}_i, \mathbf{v}\rangle\right| < \epsilon \|\mathbf{v}\|$. Noting that $\log(1 - x) < -x$ and assuming $\lambda_1 > 2\lambda_i$, we have

$$t > \lambda_1(2\lambda_i)^{-1} \log\left(|\langle\mathbf{e}_i, \mathbf{v}\rangle|\epsilon^{-1} \|\mathbf{v}\|^{-1}\right) \tag{4}$$

**The condition number.** We are primarily interested in the case when $d$ is infinite or very large and the corresponding operators/matrices are extremely ill-conditioned with infinite or approaching infinity condition number. In that case instead of a single condition number, one should consider the properties of eigenvalue decay.

**Gradient descent, smoothness and kernel methods.** We now proceed to analyze the computational reach for kernel methods. We will start by discussing the case of *infinite data* (the population case). It is both easier to analyze and allows us to demonstrate the purely computational (non-statistical) nature of limitations of gradient descent. We will see that when the kernel is smooth, the reach of gradient descent is limited to very smooth, at least infinitely differentiable functions. Moreover, to approximate a function with less smoothness within some accuracy $\epsilon$ in the $L^2$ norm one needs a super-polynomial (or even exponential) in $1/\epsilon$ number of iterations of gradient descent. Let the data be sampled from a probability with a smooth density $\mu$ on a compact domain $\Omega \subset \mathbb{R}^p$. In the case of infinite data $H$ becomes an integral operator corresponding to a positive definite kernel $k(\cdot, \cdot)$ such that $\mathcal{K}f(x) \stackrel{\text{def}}{=} \int_\Omega k(x, z)f(z)d\mu_z$. This is a compact self-adjoint operator with an infinite positive spectrum $\lambda_1, \lambda_2, \ldots$, $\lim_{i\to\infty}\lambda_i = 0$. We have (see the full paper for discussion and references):

**Theorem 1.** *If $k$ is an infinitely differentiable kernel, the rate of eigenvalue decay is super-polynomial, i.e. $\lambda_i = O(i^{-P}) \quad \forall P \in \mathbb{N}$. Moreover, if $k$ is the Gaussian kernel, there exist constants $C, C' > 0$ such that for large enough $i$, $\lambda_i < C' \exp\left(-Ci^{1/p}\right)$.*

**The computational reach of kernel methods.** Consider the eigenfunctions of $\mathcal{K}$, $\mathcal{K}e_i = \lambda_i e_i$, which form an orthonormal basis for $L^2(\Omega)$. We can write a function $f \in L^2(\Omega)$ as $f = \sum_{i=1}^\infty a_i e_i$. We have $\|f\|_{L^2}^2 = \sum_{i=1}^\infty a_i^2$. We can now describe the reach of kernel methods with smooth kernel (in the infinite data setting). Specifically, functions which can be approximated in a polynomial number of iterations must have super-polynomial coefficient decay.

**Theorem 2.** *Suppose $f \in L^2(\Omega)$ is such that it can be approximated within $\epsilon$ using a polynomial in $1/\epsilon$ number of gradient descent iterations, i.e. $\forall_{\epsilon>0} f \in \mathcal{CR}_{\epsilon^{-M}}(\epsilon)$ for some $M \in \mathbb{N}$. Then any $N \in \mathbb{N}$ and $i$ large enough $|a_i| < i^{-N}$.*

**Corollary 1.** *Any $f \in L^2(\Omega)$ which can be $\epsilon$-approximated with polynomial in $1/\epsilon$ number of steps of gradient descent is infinitely differentiable. In particular, $f$ function must belong to the intersection of all Sobolev spaces on $\Omega$.*

**Gradient descent for periodic functions on $\mathbb{R}$.** Let us now consider a simple but important special case, where the reach can be analyzed very explicitly. Let $\Omega$ be a circle with the uniform measure, or, equivalently, consider periodic functions on the interval $[0, 2\pi]$. Let $k_s(x, z)$ be the heat kernel on the circle [Ros97]. This kernel is very close to the Gaussian kernel $k_s(x, z) \approx \frac{1}{\sqrt{2\pi s}} \exp\left(-\frac{(x-z)^2}{4s}\right)$. The eigenfunctions $e_j$ of the integral operator $\mathcal{K}$ corresponding to $k_s(x, z)$ are simply the Fourier harmonics $\sin jx$ and $\cos jx$. The corresponding eigenvalues are $\{1, e^{-s}, e^{-s}, e^{-4s}, e^{-4s}, \ldots, e^{-\lfloor j/2+1\rfloor^2 s}, \ldots\}$. Given a function $f$ on $[0, 2\pi]$, we can write its Fourier series $f = \sum_{j=0}^{\infty} a_j e_j$. A direct computation shows that for any $f \in CR_t(\epsilon)$, we have $\sum_{i > \frac{\sqrt{2\ln 2t}}{s}} a_i^2 < 3\epsilon^2 \|\mathbf{v}\|^2$. We see that the space $f \in CR_t(\epsilon)$ is "frozen" as $\sqrt{2\ln 2t} s$ grows extremely slowly as the number of iterations $t$ increases. As a simple example consider the Heaviside step function $f(x)$ (on a circle), taking 1 and $-1$ values for $x \in (0, \pi]$ and $x \in (\pi, 2\pi]$, respectively. The step function can be written as $f(x) = \frac{4}{\pi} \sum_{j=1,3,\ldots} \frac{1}{j} \sin(jx)$. From the analysis above, we need $O(\exp(\frac{s}{\epsilon^2}))$ iterations of gradient descent to obtain an $\epsilon$-approximation to the function. It is important to note that the Heaviside step function is a rather natural example, especially in the classification setting, where it represents the simplest two-class classification problem. The situation is not much better for functions with more smoothness unless they happen to be extremely smooth with super-exponential Fourier component decay. In contrast, a direct computation of inner products $\langle f, e_i \rangle$ yields *exact* function recovery for any function in $L^2([0, 2\pi])$ using the amount of computation equivalent to just *one step* of gradient descent. Thus, we see that the gradient descent is an extremely inefficient way to recover Fourier series for a general periodic function. The situation is only mildly improved in dimension $d$, where the span of at most $O^*\left((\log t)^{d/2}\right)$ eigenfunctions of a Gaussian kernel or $O\left(t^{1/p}\right)$ eigenfunctions of an arbitrary $p$-differentiable kernel can be approximated in $t$ iterations. The discussion above shows that the gradient descent with a smooth kernel can be viewed as a heavy regularization of the target function. It is essentially a band-limited approximation no more than $O(\ln t)$ Fourier harmonics. While regularization is often desirable from a generalization/finite sample point of view , especially when the number of data points is small, the bias resulting from the application of the gradient descent algorithm cannot be overcome in a realistic number of iterations unless target functions are extremely smooth or the kernel itself is not infinitely differentiable.

**Remark: Rate of convergence vs statistical fit.** Note that we can improve convergence by changing the shape parameter of the kernel, i.e. making it more "peaked" (e.g., decreasing the bandwidth $s$ in the definition of the Gaussian kernel) While that does not change the exponential nature of the asymptotics of the eigenvalues, it slows their decay. Unfortunately improved convergence comes at the price of overfitting. In particular, for finite data, using a very narrow Gaussian kernel results in an approximation to the 1-NN classifier, a suboptimal method which is up to a factor of two inferior to the Bayes optimal classifier in the binary classification case asymptotically.

**Finite sample effects, regularization and early stopping.** It is well known (e.g., [B$^+$05, RBV10]) that the top eigenvalues of kernel matrices approximate the eigenvalues of the underlying integral operators. Therefore computational obstructions encountered in the infinite case persist whenever the data set is large enough. Note that for a kernel method, $t$ iterations of gradient descent for $n$ data points require $t \cdot n^2$ operations. Thus, gradient descent is computationally pointless unless $t \ll n$. That would allow us to fit only about $O(\log t)$ eigenvectors. In practice we need $t$ to be much smaller than $n$, say, $t < 1000$. At this point we should contrast our conclusions with the important analysis of early stopping for gradient descent provided in [YRC07] (see also [RWY14, CARR16]). The authors analyze gradient descent for kernel methods obtaining the optimal number of iterations of the form $t = n^\theta, \theta \in (0, 1)$. That seems to contradict our conclusion that a very large, potentially exponential, number of iterations may be needed to guarantee convergence. The apparent contradiction stems from the assumption in [YRC07] that the regression function $f^*$ belongs to the range of some power of the kernel operator $K$. For an infinitely differentiable kernel, that implies super-polynomial spectral decay ($a_i = O(\lambda_i^N)$ for any $N > 0$). In particular, it implies that $f^*$ belongs to any Sobolev space. We do not typically expect such high degree of smoothness in practice, particularly in classification problems, where the Heaviside step function seems to be a reasonable model. In particular, we expect

sharp transitions of label probabilities across class boundaries to be typical for many classifications datasets. These areas of near-discontinuity will necessarily result in slow decay of Fourier coefficients and require many iterations of gradient descent to approximate[1].

To illustrate this point, we show (right table) the results of gradient descent for two datasets of 10000 points (see Section 6). The regression error on the training set is roughly inverse to the number of iterations, i.e. every extra bit of precision requires

| Dataset | Metric | | Number of iterations | | | | |
|---|---|---|---|---|---|---|---|
| | | | 1 | 80 | 1280 | 10240 | 81920 |
| MNIST-10k | L2 loss | train | 4.07e-1 | 9.61e-2 | 2.60e-2 | 2.36e-3 | 2.17e-5 |
| | | test | 4.07e-1 | 9.74e-2 | 4.59e-2 | 3.64e-2 | **3.55e-2** |
| | c-error (test) | | 38.50% | 7.60% | 3.26% | **2.39%** | 2.49% |
| HINT-M-10k | L2 loss | train | 8.25e-2 | 4.58e-2 | 3.08e-2 | 1.83e-2 | 4.21e-3 |
| | | test | 7.98e-2 | 4.24e-2 | 3.34e-2 | **3.14e-2** | 3.42e-2 |

twice the number of iterations for the previous bit. For comparison, we see that the minimum regression ($L^2$) error on both test sets is achieved at over $10000$ iterations. This results is at least *cubic* computational complexity equivalent to that of a direct method.

**Regularization.** Note that typical regularization, e.g., adding $\lambda \|f\|$, results in discarding information along the directions with small eigenvalues (below $\lambda$). While this improves the condition number it comes at a high cost in terms of over-regularization. In the Fourier analysis example this is similar to considering band-limited functions with $\sim \sqrt{\log(1/\lambda)}/s$ Fourier components. Even for $\lambda = 10^{-16}$ (limit of double precision) and $s = 1$ we can only fit about 10 Fourier components. We argue that there is little need for explicit regularization for most iterative methods in the big data regimes.

## 3 Extending the reach of gradient descent: EigenPro iteration

We will now propose practical measures to alleviate the over-regularization of linear regression by gradient descent. As seen above, one of the key shortcomings of shallow learning methods based on smooth kernels (and their approximations, e.g., Fourier and RBF features) is their fast spectral decay. That suggests modifying the corresponding matrix $H$ by decreasing its top eigenvalues, enabling the algorithm to approximate more target functions in the same number of iterations. Moreover, this can be done in a way compatible with stochastic gradient descent thus obviating the need to materialize full covariance/kernel matrices in memory. Accurate approximation of top eigenvectors can be obtained from a subsample of the data with modest computational expenditure. Combining these observations we propose EigenPro, a low overhead preconditioned Richardson iteration.

**Preconditioned (stochastic) gradient descent.** We will modify the linear system in Eq. 1 with an invertible matrix $P$, called a left preconditioner. $PH\alpha - P\mathbf{b} = 0$. Clearly, this modified system and the original system in Eq. 1 have the same solution. The Richardson iteration corresponding to the modified system (preconditioned Richardson iteration) is

$$\alpha^{(t+1)} = \alpha^{(t)} - \eta P(H\alpha^{(t)} - \mathbf{b}) \tag{5}$$

It is easy to see that as long as $\eta \|PH\| < 1$ it converges to $\alpha^*$, the solution of the original linear system. Preconditioned SGD can be defined similarly by

$$\alpha \leftarrow \alpha - \eta P(H_m \alpha - \mathbf{b}_m) \tag{6}$$

where we define $H_m \overset{\text{def}}{=} \frac{1}{m} X_m^T X_m$ and $b_m \overset{\text{def}}{=} \frac{1}{m} X_m^T \mathbf{y}_m$ using sampled mini-batch $(X_m, \mathbf{y}_m)$.

**Preconditioning as a linear feature map.** It is easy to see that the preconditioned iteration is in fact equivalent to the standard Richardson iteration in Eq. 2 on a dataset transformed with the linear feature map, $\phi_P(\mathbf{x}) \overset{\text{def}}{=} P^{\frac{1}{2}} \mathbf{x}$. This is a convenient point of view as the transformed data can be stored for future use. It also shows that preconditioning is compatible with most computational methods both in practice and, potentially, in terms of analysis.

**Linear EigenPro.** We will now discuss properties desired to make preconditioned GD/SGD methods effective on large scale problems. Thus for the modified iteration in Eq. 5 we would like to choose $P$ to meet the following targets: (Acceleration) The algorithm should provide high accuracy in a small number of iterations. (Initial cost) The preconditioning matrix $P$ should be accurately computable, without materializing the full covariance matrix. (Cost per iteration) Preconditioning by $P$ should be efficient per iteration in terms of computation and memory. The convergence of the preconditioned algorithm with the along the $i$-th eigendirection is dependent on the ratio of eigenvalues $\lambda_i(PH)/\lambda_1(PH)$. This leads us to choose the preconditioner $P$ to maximize the ratio $\lambda_i(PH)/\lambda_1(PH)$ for each $i$. We see that modifying the top eigenvalues of $H$ makes the most difference in convergence. For example, decreasing $\lambda_1$ improves convergence along all directions, while decreasing any other eigenvalue only speeds up convergence in that

direction. However, decreasing $\lambda_1$ below $\lambda_2$ does not help unless $\lambda_2$ is decreased as well. Therefore it is natural to decrease the top $k$ eigenvalues to the maximum amount, i.e. to $\lambda_{k+1}$, leading to

$$P \overset{\text{def}}{=} I - \sum_{i=1}^{k} (1 - \lambda_{k+1}/\lambda_i)\mathbf{e}_i\mathbf{e}_i^T \quad (7)$$

We see that $P$-preconditioned iteration increases convergence by a factor $\lambda_1/\lambda_k$. However, exact construction of $P$ involves computing the eigendecomposition of the $d \times d$ matrix $H$, which is not feasible for large data. Instead we use subsampled randomized SVD [HMT11] to obtain an approximate preconditioner $\hat{P}_\tau \overset{\text{def}}{=} I - \sum_{i=1}^{k} (1 - \tau\hat{\lambda}_{k+1}/\hat{\lambda}_i)\hat{\mathbf{e}}_i\hat{\mathbf{e}}_i^T$. Here algorithm RSVD (detailed in the full paper) computes the approximate top eigenvectors $E \leftarrow (\hat{\mathbf{e}}_1, \ldots, \hat{\mathbf{e}}_k)$ and eigenvalues $\Lambda \leftarrow \text{diag}(\hat{\lambda}_1, \ldots, \hat{\lambda}_k)$ and

**Algorithm:** $\text{EigenPro}(X, \mathbf{y}, k, m, \eta, \tau, M)$
**input** training data $(X, \mathbf{y})$, number of eigendirections $k$, mini-batch size $m$, step size $\eta$, damping factor $\tau$, subsample size $M$
**output** weight of the linear model $\alpha$
1: $[E, \Lambda, \hat{\lambda}_{k+1}] = \text{RSVD}(X, k+1, M)$
2: $P \overset{\text{def}}{=} I - E(I - \tau\hat{\lambda}_{k+1}\Lambda^{-1})E^T$
3: Initialize $\alpha \leftarrow 0$
4: **while** stopping criteria is False **do**
5: $\quad (X_m, \mathbf{y}_m) \leftarrow m$ rows sampled from $(X, \mathbf{y})$ without replacement
6: $\quad \mathbf{g} \leftarrow \frac{1}{m}(X_m^T(X_m\alpha) - X_m^T\mathbf{y}_m)$
7: $\quad \alpha \leftarrow \alpha - \eta P\mathbf{g}$
8: **end while**

$\hat{\lambda}_{k+1}$ for subsample covariance matrix $H_M$. We introduce the parameter $\tau$ to counter the effect of approximate top eigenvectors "spilling" into the span of the remaining eigensystem. Using $\tau < 1$ is preferable to the obvious alternative of decreasing the step size $\eta$ as it does not decrease the step size in the directions nearly orthogonal to the span of $(\hat{\mathbf{e}}_1, \ldots, \hat{\mathbf{e}}_k)$. That allows the iteration to converge faster in those directions. In particular, when $(\hat{\mathbf{e}}_1, \ldots, \hat{\mathbf{e}}_k)$ are computed exactly, the step size in other eigendirections will not be affected by the choice of $\tau$. We call SGD with the preconditioner $\hat{P}_\tau$ (Eq. 6) *EigenPro iteration*. See Algorithm EigenPro for details. Moreover, the key step size parameter $\eta$ can be selected in a theoretically sound way discussed below.

**Kernel EigenPro.** We will now discuss modifications needed to work directly in the RKHS (primal) setting. A positive definite kernel $\text{k}(\cdot, \cdot) : \mathbb{R}^N \times \mathbb{R}^N \to \mathbb{R}$ implies a feature map from $X$ to an RKHS space $\mathcal{H}$. The feature map can be written as $\phi : x \mapsto k(x, \cdot), \mathbb{R}^N \to \mathcal{H}$. This feature map leads to the learning problem $f^* = \arg\min_{f \in \mathcal{H}} \frac{1}{n}\sum_{i=1}^{n}(\langle f, \text{k}(\mathbf{x}_i, \cdot)\rangle_{\mathcal{H}} - y_i)^2$. Using properties of RKHS, EigenPro iteration in $\mathcal{H}$ becomes $f \leftarrow f - \eta\,\text{P}(\mathcal{K}(f) - \text{b})$ where $\text{b} \overset{\text{def}}{=} \frac{1}{n}\sum_{i=1}^{n} y_i k(\mathbf{x}_i, \cdot)$ and covariance operator $\mathcal{K} = \frac{1}{n}\sum_{i=1}^{n} \text{k}(\mathbf{x}_i, \cdot) \otimes \text{k}(\mathbf{x}_i, \cdot)$. The top eigensystem of $\mathcal{K}$ forms the preconditioner $\text{P} \overset{\text{def}}{=} \text{I} - \sum_{i=1}^{k}(1 - \tau\lambda_{k+1}(\mathcal{K})/\lambda_i(\mathcal{K}))\,\text{e}_i(\mathcal{K}) \otimes \text{e}_i(\mathcal{K})$. By the Representer theorem [Aro50], $f^*$ admits a representation of the form $\sum_{i=1}^{n}\alpha_i\,\text{k}(\mathbf{x}_i, \cdot)$. Parameterizing the above iteration accordingly and applying some linear algebra lead to the following iteration in a finite-dimensional vector space, $\alpha \leftarrow \alpha - \eta P(K\alpha - \mathbf{y})$ where $K \overset{\text{def}}{=} [\text{k}(\mathbf{x}_i, \mathbf{x}_j)]_{i,j=1,\ldots,n}$ is the kernel matrix and EigenPro preconditioner $P$ is defined using the top eigensystem of $K$ (assume $K\mathbf{e}_i = \lambda_i\mathbf{e}_i$), $P \overset{\text{def}}{=} I - \sum_{i=1}^{k}\lambda_i^{-1}(1 - \tau\lambda_{k+1}/\lambda_i)\mathbf{e}_i\mathbf{e}_i^T$. This differs from that for the linear case (Eq. 7) (with an extra factor of $1/\lambda_i$) due to the difference between the parameter space of $\alpha$ and the RKHS space.

**EigenPro as kernel learning.** Another way to view EigenPro is in terms of kernel learning. Assuming that the preconditioner is computed exactly, EigenPro is equivalent to computing the (distribution-dependent) kernel, $k_{EP}(x, z) \overset{\text{def}}{=} \sum_{i=1}^{k}\lambda_{k+1}e_i(x)e_i(z) + \sum_{i=k+1}^{\infty}\lambda_i e_i(x)e_i(z)$. Notice that the RKHS spaces corresponding to $k_{EP}$ and $k$ contain the same functions but have different norms. The norm in $k_{EP}$ is a finite rank modification of the norm in the RKHS corresponding to $k$, a setting reminiscent of [SNB05] where unlabeled data was used to "warp" the norm for semi-supervised learning. However, in our paper the "warping" is purely for computational efficiency.

**Acceleration.** EigenPro can obtain acceleration factor of up to $\frac{\lambda_1}{\lambda_{k+1}}$ over the standard gradient descent. That factor assumes full gradient descent and exact computation of the preconditioner. See below for an acceleration analysis in the SGD setting.

**Initial cost.** To construct the preconditioner $P$, we perform RSVD to compute the approximate top eigensystem of covariance $H$. RSVD has time complexity $O(Md\log k + (M+d)k^2)$ (see [HMT11]). The subsample size $M$ can be much smaller than the data size $n$ while preserving the accuracy of estimation. In addition, extra $kd$ memory is needed to store the eigenvectors.

**Cost per iteration.** For standard SGD using $d$ kernel centers (or random Fourier features) and mini-batch of size $m$, the computational cost per iteration is $O(md)$. In comparison, EigenPro iteration using top-$k$ eigen-directions costs $O(md + kd)$. Specifically, applying preconditioner $P$ in EigenPro requires left multiplication by a matrix of rank $k$. This involves $k$ vector-vector dot products resulting in $k \cdot d$ additional operations per iteration. These can be implemented efficiently on a GPU.

# 4 Step Size Selection for EigenPro Preconditioned Methods

We will now discuss the key issue of the step size selection for EigenPro iteration. For iteration involving covariance matrix $H$, $\lambda_1(H)^{-1} = \|H\|^{-1}$ results in optimal (within a factor of 2) convergence. This suggests choosing the corresponding step size $\eta = \|PH\|^{-1} = \lambda_{k+1}^{-1}$. In practice this will lead to divergence due to (1) approximate computation of eigenvectors (2) the randomness inherent in SGD. One (costly) possibility is to compute $\|PH_m\|$ at every step. As the mini-batch can be assumed to be chosen at random, we propose using a lower bound on $\|H_m\|^{-1}$ (with high probability) as the step size to guarantee convergence at each iteration.

**Linear EigenPro.** Consider the EigenPro preconditioned SGD in Eq. 6. For this analysis assume that $P$ is formed by the exact eigenvectors. Interpreting $P^{\frac{1}{2}}$ as a linear feature map as in Section 2, makes $P^{\frac{1}{2}}H_m P^{\frac{1}{2}}$ a random subsample on the dataset $XP^{\frac{1}{2}}$. Using matrix Bernstein [Tro15] yields

**Theorem 3.** *If* $\|\mathbf{x}\|_2^2 \leq \kappa$ *for any* $\mathbf{x} \in X$ *and* $\lambda_{k+1} = \lambda_{k+1}(H)$*, with probability at least* $1 - \delta$, $\|PH_m\| \leq \lambda_{k+1} + 2(\lambda_{k+1} + \kappa)(3m)^{-1}(\ln 2d\delta^{-1}) + \sqrt{2\lambda_{k+1}\kappa m^{-1}(\ln 2d\delta^{-1})}$.

**Kernel EigenPro.** For EigenPro iteration in RKHS, we can bound $\|P \circ \mathcal{K}_m\|$ with a very similar result based on operator Bernstein [Min17]. Note that dimension $d$ in Theorem 3 is replaced by the intrinsic dimension [Tro15]. See the arXiv version of this paper for details.

**Choice of the step size.** In the spectral norm bounds $\lambda_{k+1}$ is the dominant term when the mini-batch size $m$ is large. However, in most large-scale settings, $m$ is small, and $\sqrt{2\lambda_{k+1}\kappa/m}$ becomes the dominant term. This suggests choosing step size $\eta \sim 1/\sqrt{\lambda_{k+1}}$ leading to acceleration on the order of $\lambda_1/\sqrt{\lambda_{k+1}}$ over the standard (unpreconditioned) SGD. That choice works well in practice.

# 5 EigenPro and Related Work

Large scale machine learning imposes fairly specific limitations on optimization methods. The computational budget allocated to the problem must not exceed $O(n^2)$ operations, a small number of matrix-vector multiplications. That rules out most direct second order methods which require $O(n^3)$ operations. Approximate second order methods are far more efficient. However, they typically rely on low rank matrix approximation, a strategy which (similarly to regularization) in conjunction with smooth kernels discards information along important eigen-directions with small eigenvalues. On the other hand, first order methods can be slow to converge along eigenvectors with small eigenvalues. An effective method must thus be a hybrid approach using approximate second order information in a first order method. EigenPro is an example of such an approach as the second order information is used in conjunction with a first order method. The things that make EigenPro effective are as follows:
1. The second order information (eigenvalues and eigenvectors) is computed efficiently from a subsample of the data. Due to the quadratic loss function, that computation needs to be conducted only once. Moreover, the step size can be fixed throughout the iterations.
2. Preconditioning by a low rank modification of the identity matrix results in low overhead per iteration. The update is computed without materializing the full preconditioned covariance matrix.
3. EigenPro iteration converges (mathematically) to the same result even if the second order approximation is not accurate. That makes EigenPro relatively robust to errors in the second order preconditioning term $P$, in contrast to most approximate second order methods.

**Related work: First order optimization methods.** Gradient based methods, such as gradient descent (GD), stochastic gradient descent (SGD), are classical methods [She94, DJS96, BV04, Bis06]. Recent success of neural networks had drawn significant attention to improving and accelerating these methods. Methods like SAGA [RSB12] and SVRG [JZ13] improve stochastic gradient by periodically evaluating full gradient to achieve variance reduction. Algorithms in [DHS11, TH12, KB14] compute adaptive step size for each gradient coordinate.

**Scalable kernel methods.** There is a significant literature on scalable kernel methods including [KSW04, HCL+08, SSSSC11, TBRS13, DXH+14] Most of these are first order optimization methods. To avoid the $O(n^2)$ computation and memory requirement typically involved in constructing the kernel matrix, they often adopt approximations like RBF features [WS01, QB16, TRVR16] or random Fourier features [RR07, LSS13, DXH+14, TRVR16].

**Second order/hybrid optimization methods.** Second order methods use the inverse of the Hessian matrix or its approximation to accelerate convergence [SYG07, BBG09, MNJ16, BHNS16, ABH16]. These methods often need to compute the full gradient every iteration [LN89, EM15, ABH16] making less suitable for large data. [EM15] analyzed a hybrid first/second order method for general convex optimization with a rescaling term based on the top eigenvectors of the Hessian. That can be viewed as preconditioning the Hessian at every GD iteration. A related recent work [GOSS16]

analyses a hybrid method designed to accelerate SGD convergence for ridge regression. The data are preprocessed by rescaling points along the top singular vectors of the data matrix. Another second order method PCG [ACW16] accelerates the convergence of conjugate gradient for large kernel ridge regression using a preconditioner which is the inverse of an approximate covariance generated with random Fourier features. [TRVR16] achieves similar preconditioning effects by solving a linear system involving a subsampled kernel matrix every iteration. While not strictly a preconditioner Nyström with gradient descent(NYTRO) [CARR16] also improves the condition number. Compared to many of these methods EigenPro directly addresses the underlying issues of slow convergence without introducing a bias in directions with small eigenvalues. Additionally EigenPro incurs only a small overhead per iteration both in memory and computation.

## 6  Experimental Results

**Computing Resource/Data/Metrics.** Experiments were run on a workstation with 128GB main memory, two Intel Xeon(R) E5-2620 CPUs, and one GTX Titan X (Maxwell) GPU. For multiclass datasets, we report classification error (**c-error**) for binary valued labels and mean squared error (**mse**) for real valued labels. See the arXiv version for details and more experimental results.

**Kernel methods/Hyperparameters.** For smaller datasets direct solution of kernel regularized least squares (**KRLS**) is used to obtain the reference error. We compare with the primal method Pegasos [SSSSC11]. For even larger datasets, we use Random Fourier Features [RR07] (**RF**) with SGD as in [DXH$^+$14, TRVR16]. The results of these methods are presented as baselines. For consistent comparison, all iterative methods use mini-batch of size $m = 256$. EigenPro preconditioner is constructed using the top $k = 160$ eigenvectors of a subsampled dataset of size $M = 4800$. For EigenPro-RF, we set the damping factor $\tau = 1/4$. For primal EigenPro $\tau = 1$.

**Acceleration for different kernels.** The table on the right presents the number of epochs needed by EigenPro and Pegasos to reach the error of the optimal kernel classifier. We see that EigenPro provides acceleration of 6 to 35 times in terms of the

| Dataset | Size | Gaussian | | Laplace | | Cauchy | |
|---|---|---|---|---|---|---|---|
| | | EigPro | Pega | EigPro | Pega | EigPro | Pega |
| MNIST | $6 \cdot 10^4$ | **7** | 77 | **4** | 143 | **7** | 78 |
| CIFAR-10 | $5 \cdot 10^4$ | **5** | 56 | **13** | 136 | **6** | 107 |
| SVHN | $7 \cdot 10^4$ | **8** | 54 | **14** | 297 | **17** | 191 |
| HINT-S | $5 \cdot 10^4$ | **19** | 164 | **15** | 308 | **13** | 126 |

number of epochs required without any loss of accuracy. The actual acceleration is about $20\%$ less due to the overhead of maintaining and applying a preconditioner.

**Comparisons on large datasets**. Table below compares EigenPro to Pegasos/SGD-RF on several large datasets for 10 epochs. We see that EigenPro consistently outperforms Pegasos/SGD-RF within a fixed computational budget. Note that we adopt Gaussian kernel and $2 \cdot 10^5$ random features.

| Dataset | Size | Metric | EigenPro | | Pegasos | | EigenPro-RF | | SGD-RF | |
|---|---|---|---|---|---|---|---|---|---|---|
| | | | result | GPU hours | result | GPU hours | result | GPU hours | result | GPU hours |
| HINT-S | $2 \cdot 10^5$ | c-error | **10.0%** | 0.1 | 11.7% | 0.1 | 10.3% | 0.2 | 11.5% | 0.1 |
| TIMIT | $1 \cdot 10^6$ | | **31.7%** | 3.2 | 33.0% | 2.2 | 32.6% | 1.5 | 33.3% | 1.0 |
| MNIST-8M | $1 \cdot 10^6$ | | **0.8%** | 3.0 | 1.1% | 2.7 | 0.8% | 0.8 | 1.0% | 0.7 |
| | $8 \cdot 10^6$ | | - | | - | | **0.7%** | 7.2 | 0.8% | 6.0 |
| HINT-M | $1 \cdot 10^6$ | mse | **2.3e-2** | 1.9 | 2.7e-2 | 1.5 | 2.4e-2 | 0.8 | 2.7e-2 | 0.6 |
| | $7 \cdot 10^6$ | | - | | - | | **2.1e-2** | 5.8 | 2.4e-2 | 4.1 |

**Comparisons to state-of-the-art**. In the below table, we provide a comparison to several large scale kernel results reported in the literature. EigenPro improves or matches performance on each dataset at a much lower computational budget. We note that [MGL$^+$17] achieves error 30.9% on TIMIT using an AWS cluster. The method uses a novel supervised feature selection method, hence is not directly comparable. EigenPro can plausibly further improve the training error using this new feature set.

| Dataset | Size | EigenPro (use 1 GTX Titan X) | | | Reported results | | |
|---|---|---|---|---|---|---|---|
| | | error | GPU hours | epochs | source | error | description |
| MNIST | $1 \cdot 10^6$ | **0.70%** | 4.8 | 16 | [ACW16] | 0.72% | 1.1 hours/189 epochs/1344 AWS vCPUs |
| | $6.7 \cdot 10^6$ | **0.80%**$^\dagger$ | 0.8 | 10 | [LML$^+$14] | 0.85% | less than 37.5 hours on 1 Tesla K20m |
| TIMIT | $2 \cdot 10^6$ | **31.7%** (32.5%)$^\ddagger$ | 3.2 | 10 | [HAS$^+$14] | 33.5% | 512 IBM BlueGene/Q cores |
| | | | | | [TRVR16] | 33.5% | 7.5 hours on 1024 AWS vCPUs |
| SUSY | $4 \cdot 10^6$ | **19.8%** | 0.1 | 0.6 | [CAS16] | $\approx 20\%$ | 0.6 hours on IBM POWER8 |

$\dagger$ The result is produced by EigenPro-RF using $1 \times 10^6$ data points.    $\ddagger$ Our TIMIT training set ($1 \times 10^6$ data points) was generated following a standard practice in the speech community [PGB$^+$11] by taking 10ms frames and dropping the glottal stop 'q' labeled frames in core test set (1.2% of total test set). [HAS$^+$14] adopts 5ms frames, resulting in $2 \times 10^6$ data points, and keeping the glottal stop 'q'. In the worst case scenario EigenPro, if we mislabel all glottal stops, the corresponding frame-level error increases from 31.7% to 32.5%.

**Acknowledgements.** We thank Adam Stiff, Eric Fosler-Lussier, Jitong Chen, and Deliang Wang for providing TIMIT and HINT datasets. This work is supported by NSF IIS-1550757 and NSF CCF-1422830. Part of this work was completed while the second author was at the Simons Institute at Berkeley. In particular, he thanks Suvrit Sra, Daniel Hsu, Peter Bartlett, and Stefanie Jegelka for many discussions and helpful suggestions.

## Footnotes

[1] Interestingly they can lead to lower sample complexity for optimal classifiers (cf. Tsybakov margin condition [Tsy04]).

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
