[Reviews · NeurIPS 2017]

Reviewer 1



The paper studies the effect of decay of eigenvalues on function approximation using Gradient Descent(GD). It argues that Gradient Descent machinery may lead to large number of iterations when infinitely diff kernels are used to approximate non-smooth functions. To remedy this one could use second order methods or use an explicit regularisation. While explicit Regularisation leads to bias the alternative of deploying second order methods are too expensive. To remedy the situation the paper proposes Eigen-Pro. The result showing that GD is ineffective when one wishes to approximate a not so smooth function using an infinitely differentiable kernel is novel and insightful. It is linked to the decay of eigen structure of the kernel functions is again insightful. The use of eigen-pro which achieves a sort of implicit regularisation is an interesting alternative where the computational challenge is finessed with randomisation which should have practical significance While the above mentioned paragraph mentions the pluses there are two negative points that needs to be mentioned. Firstly, the implications of the derived results are not clear. If it is speed-up we are concerned with then the comparisons with PEGASOS Is only marginal and often Eigen-pro is slower. It is commendable that the method can match performance with state of the art algorithms with kernel methods more needs to be done to have an explicit comparison to understand time and accuracy trade-offs. Would it make sense to show results by training a Deep Network(since this is what is used as the state of the art ) and comparing it with the kernel approximation in a synthetic setting. This would strengthen the understanding of time and accuracy trade-offs. Secondly, the longer version of the paper is easy to read but the submitted version is not very easy to read. Overall, a nice paper, but lacking in explaining the experimental results and readability.

Reviewer 2



The paper presents an analysis of gradient descent as learning strategy for kernel methods. The main result of the paper shows that there is a restriction on the functions that can be effectively approximated by gradient descent. In particular, the paper defines the concept of 'computational reach' of gradient descent and shows that for smooth kernels the 'computational reach' includes only a small fraction of the function space. This limitation is overcame by a new method called "EigenPro" which uses a precondition strategy. The method is efficiently implemented and evaluated over standard datasets. The results show systematic improvements by the proposed algorithm, however the statistical significance of these differences is not evaluated. In general, the papers is well written, the ideas are clearly presented, and the experimental setup, as well as the results are convincing. Even though I didn't check the details of the mathematical claims, they seem to be sound. Overall, I think the work contributes important insights on how large-scale kernel learning works and how it could be improved.

Reviewer 3



This paper looks at analyzing the problem of long times required for convergence of kernel methods when optimized via gradient descent. They define a notion of computational reach of gradient descent after t iterations and the failure of gradient descent to reach the epsilon neighborhood of an optimum after t iterations. They give examples of simplistic function settings with binary labels where gradient descent takes a long time to converge to the optimum. They also point out that adding regularization improves the condition number of the eigenspectrum (resulting in possible better convergence) but also leads to overregularization at times. They introduce the notion of EigenPro, where they pre-multiply the data using a preconditioner matrix, which can be pre-computed, improves the time for convergence by making the lower eigenvalues closer to the largest one as well as making cost per iteration efficient. They do a randomized SVD of the data matrix/kernel operator to get the eigenvalues and generate the pre-conditioning matrix using the ratio of the eigenvalues.They show that the per-iteration time is not much higher than kernel methods and demonstrate experiments to show that the errors are minimized better than standard kernel methods using less GPU time overall. The acceleration provided is significant for some of the Kernels. The paper is dense in presentation but is well written and not very difficult to follow. However there would be a lot of details that can be provided to compare how the pre-conditioning matrix can influence gradient descent in general and whether it should always be applied to data matrices every time we try to train a linear model on data. The authors also provide the argument of how lowering the ratio of the smaller eigenvalues compared to the larger one makes the problem more amenable to convergence. It would be good to see some motivation/geometric description of how the method provides better convergence using gradient descent. It would also be interesting to explore if other faster algorithms including proximal methods as well as momentum based methods also can benefit from such pre-conditioning and can improve the rates of convergence for kernel methods.